# Comparison of the Post-Surgical Position of the Temporomandibular Joint after Orthognathic Surgery in Skeletal Class III Patients and Patients with Cleft Lip and Palate

**DOI:** 10.3390/jpm12091437

**Published:** 2022-08-31

**Authors:** Yi-Hao Lee, Chi-Yu Tsai, Ling-Chun Wang, U-Kei Lai, Jui-Pin Lai, Shiu-Shiung Lin, Yu-Jen Chang

**Affiliations:** 1Department of Craniofacial Orthodontics, Department of Dentistry, Kaohsiung Chang Gung Memorial Hospital and Chang Gung University College of Medicine, Kaohsiung 833, Taiwan; 2Department of Plastic and Reconstructive Surgery, Kaohsiung Chang Gung Memorial Hospital and Chang Gung University College of Medicine, Kaohsiung 833, Taiwan

**Keywords:** orthognathic surgery, computed tomography, temporomandibular joint stability, cleft lip and palate

## Abstract

Objective: The purpose of our research is to compare the post-surgical position of the temporomandibular joint in skeletal Class III patients and patients with cleft lip and palate treated with two-jaw orthognathic surgery using a three-dimensional computer tomography image. Materials and Methods: Twenty-three skeletal Class III patients with mandibular prognathism associated with maxillary retrognathism in group 1 and twenty cleft mid-face retrusion skeletal Class III patients in group 2 were enrolled in this study. All subjects were treated with two-jaw orthognathic surgery. Computed tomography scans were taken in all subjects at 3 weeks preoperatively and 6 months postoperatively. Three-dimensional craniofacial skeletal structures were build-up, and assessed the temporomandibular joint position changes before and after surgery. Results: Forty-three selected patients were separated into two groups. The mean age of patients was 22.39 ± 4.8 years in group 1 and 20.25 ± 3.8 years in group 2. The range of mean three-dimensional discrepancy of the selected condylar points was 0.95–1.23 mm in group 1 and 2.37–2.86 mm in group 2. The mean alteration of intercondylar angle was 2.33 ± 1.34° in group 1 and 6.30 ± 2.22° in group 2. The significant differences in the discrepancy of TMJ and changes in intercondylar angle were confirmed within the intra-group and between the two groups. Conclusions: Significant changes in postoperative TMJ position were present in both groups. Furthermore, the cleft group presented significantly more postoperative discrepancy of TMJ and more changes in intercondylar angle after surgery. This finding may be a reason leading to greater postoperative instability in cleft patients compared with skeletal Class III non-cleft patients. Clinical Trial Registration Number: IRB No: 202201108B0.

## 1. Introduction

The stability of the temporomandibular joint (TMJ) after bimaxillary orthognathic surgery (OGS) is considered an important factor affecting the relapse rate due to the adaptation mechanism after mandibular osteotomies that leads to changes in loading distribution [1]. It may result in condylar structural changes, such as condyle remodeling and resorption. The former is a physiological process, and the latter is a pathological process. Clinical symptoms of TMJ and relapse of surgical outcome may follow condyle resorption [2,3]. Alteration of TMJ position after mandibular osteotomies is another influencing factor affecting postoperative instability [1]. A previous study demonstrated that several complications following OGS, such as indirect trauma by manipulation during surgery and intra-articular edema, might contribute to the alteration of condyle position [4]. Therefore, the issue of postoperative changes in TMJ position cannot be ignored.

Cleft patients often present with more severe jawbone discrepancy than non-cleft patients, which can usually be resolved through bimaxillary OGS. There have been some studies discussing postoperative stability after OGS in cleft patients [5,6,7,8,9,10]; however, most of these studies have focused on the issue of maxillary stability and shown evidence of moderate to high relapse rates in the horizontal and vertical directions due to insufficient mid-face resolution by Le’Fort I osteotomy [11]. Park et al. (2017) demonstrated that significant counterclockwise rotation of the distal segment of the mandible causes mandibular relapse after OGS in cleft patients [10]. A previous study also reported alterations in condyle position after OGS, which often occurs after mandibular osteotomies associated with postoperative stability [1]. However, the postoperative stability of the TMJ in cleft patients remains poorly understood, and few studies have compared the condylar position change after OGS in cleft and non-cleft patients.

We used three-dimensional (3D) imaging software to reconstruct the craniofacial skeletal structure from pre- and postoperative computed tomography (CT) imagery in this study. Superimposition of the two-stage 3D image and quantitative measurements were performed. This study aimed to compare the post-surgical position of the TMJ in skeletal Class III patients and patients with cleft lip and palate treated with two-jaw OGS.

## 2. Materials and Methods

Three-dimensional imaging data were gathered from subjects and divided into two groups in this retrospective study; group 1 consisted of 23 non-growing skeletal Class III patients with mandibular prognathism associated with maxillary retrognathism, and group 2 consisted of 20 non-growing skeletal Class III cleft patients with a retrusive mid-face. All subjects underwent maxillary Le-Fort I osteotomy combined with a mandibular bilateral sagittal split osteotomy (BSSO) between January 2018 and December 2020 at the Craniofacial Center, Kaohsiung Chang Gung Memorial Hospital. Patients with degenerative TMJ disease, deformity secondary to trauma, or any systemic disease were excluded from the study. After dental decompensation by orthodontic treatment, all patients underwent surgery performed by the same surgeon, Dr. Jui-Pin Lai.

For all patients enrolled in our study, once the required preoperative orthodontic movements had been prepared, CT scans (Toshiba Aquilion 64:120 kVp, 350 mA, rotation time: 0.5 s, 64 0.5-mm slices) over the craniofacial skeletal structures were obtained 3 weeks before surgery (T1). Surgical simulation and navigation were performed using the CASNOS protocol published by Chang et al. (2017) [12]. During surgery, the maxilla and mandible were adjusted to the purposed position using the occlusal stent. The maxillomandibular complex was repositioned based on navigation and fixed to the basal bone using prefabricated miniplates, which were manufactured following the procedure presented in a previous study [12]. The internal fixation was performed using miniature titanium bone plates and cortical screws. Intermaxillary fixation stabilized the post-surgical structure by combining a fixed orthodontic appliance and supplementary elastics for 2 weeks after surgery. A second CT scan was obtained six months postoperatively (T2) to assess treatment outcomes. Depending on these two-stage CT images, the changes in TMJ position after surgery were segmented, superimposed, and quantified by two open-source software programs, ITK-SNAP (available at: http://www.itksnap.org/pmwiki/pmwiki.php) (version 2.4.0) and 3D-Slicer (available at: http://www.slicer.org) (version 4.4.0: 4 September 2015). Open-source software tools have been utilized to measure dental and skeletal changes and have been validated for intra- and inter-rater reliability [12]. 

All selected landmarks were identified and measured using CT images by the same investigator. The adjusted subjects’ head orientation was parallel to the Frankfort horizontal (FH) line connected by the Porion (Po) and Orbitale (Or). Using the FH line as the horizontal reference, the FH plane was constructed with three points: a landmark of the middle point of both the Porion (mid-Po) and the bilateral Orbitle (Figure 1). These landmarks were identified on T1 (three weeks before surgery) and T2 (six months after surgery) scans. The superimposition of the T1 and T2 scans was registered to the cranial base using a voxel-based registration algorithm (Figure 2) [12,13].

The anatomical landmarks of the TMJ in the two groups were established and identified along the axis of the FH plane as preoperative (T1): RL (right), LL (left): the most lateral point of the condyle, RM (right), LM (left): the most medial point of the condyle; and postoperative (T2): RL’ (right), LL’ (left): the most lateral point of the condyle, RM’ (right), LM’ (left): the most medial point of the condyle (Figure 3: preoperative and Figure 4: postoperative) [14]. 

All corresponding 3D points were visualized using the 3D Slicer’s (version 4.4.0: http://www.slicer.org) quantitative 3D cephalometric (quantification of 3D components [Q3DC]) tool [12,13]. The tool permits users to study the 3D distance and alteration of the 3D angle along each of the axes between T1 and registered T2 TMJ by the placement of fiducial markers. Subsequently, the 3D distance representing the discrepancy of the condylar points between the most medial point of the condyles (RM-RM’, LM-LM’) and the most lateral point of the condyles (RL-RL’, LL-LL’) in T1 and T2 imaging was surveyed. In addition, the cutting angle between the axes (intercondylar angle) and alteration of the intercondylar angle was measured and calculated [14]. 

### Statistical Analysis

The Wilcoxon test was used to distinguish the differences in the intercondylar angle before and after surgery. Moreover, comparison of the distance of target landmarks and alteration of the intercondylar angle in the two groups preoperatively and postoperatively were calculated using the Mann–Whitney U test. The significance level was set at *p* < 0.05.

Intra-rater reliability was measured by the same researcher using intraclass correlations for three variables, including two 3D distances and one cutting angle in 10 randomly selected subjects. Repeated measurements were taken for each subject two weeks apart for the error test. No statistical differences were observed in defining the points and angles among the 3D quantitative points.

## 3. Results

Forty-three patients were divided into two groups: Group 1: skeletal Class III patients; Group 2: skeletal Class III cleft patients with a retrusive mid-face who underwent bimaxillary surgery treatment, demanding Le Fort I maxillary advancement and BSSO setback of the mandible. In group 1, 23 patients were aged between 19 and 36 years (mean age 24.39 ± 4.8 years). Twelve of these were women (mean age 23.92 ± 5.2; range 19–36 years), and 11 were men (mean age 24.9 ± 4.5, ranging from 20 to 33 years). In group 2, 20 patients were aged between 18 and 22 years (mean age 20.25 ± 3.8 years). Six of these were women (mean age 19.2 ± 2.3, ranging from 19 to 22 years), and 14 were men (mean age 19.5 ± 2.5; range, 18–21 years) (Table 1). All subjects in both groups were diagnosed with skeletal class III and maxillary deficiency. Cephalometric measurements of subjects three weeks before and 2–3 days after OGS were analyzed first. Initially, the average ANB was −6.2 ± 1.9 in group 1 and −4.9 ± 3.1 in group 2. The mean preoperative distances from point A to the N-perpendicular line (A-Nv) and Pogonion to the N-perpendicular line (Pog-Nv) were 0.5 ± 1.6 and 10.7 ± 3.7 in group 1 and −7.1 ± 4.4 and −2.9 ± 9.8 in group 2, respectively. After two-jaw surgery, the mean ANB (2.3 ± 1.5 in group 1; 0.9 ± 2.6 in group 2) was improved. The mean A-Nv (2.5 ± 1.2 in group 1; −2.6 ± 4.2 in group −2) and Pog-Nv (1.3 ± 0.6 in group 1; −5.9 ± 7.5 in group 2) also showed significant improvement in both groups postoperatively (Table 2).

Table 3 presents the discrepancy in the most lateral medial condylar points between the two groups. In group 1, the mean 3D discrepancy of the right most lateral condylar point (RL-RL’) was 1.23 ± 0.47 mm, and the mean discrepancy of the left most lateral condylar point (LL-LL’) was 1.14 ± 0.33 mm. The mean discrepancy of the right most medial condylar point (RM-RM’) was 1.09 ± 0.23 mm, and that of the left most medial condylar point (LM-LM’) was 0.95 ± 0.15 mm. The mean 3D discrepancy of the RL-RL’, LL-LL’, RM-RM’, and LM-LM’ were 2.53 ± 0.82 mm, 2.37 ± 0.71 mm, 2.86 ± 1.02 mm, and 2.42 ± 0.85 mm, respectively, in group 2. The 3D discrepancy of all parameters was significantly different between the two groups, as confirmed by the Mann–Whitney U test (*p* < 0.001).

Table 4 shows the variation and alteration in the intercondylar angles between the two groups. The angle between the condyles was measured at the crossing of lines along the longitudinal axis of the condyle. The mean angle in group 1 was 161.75 ± 5.18° before and 159.36 ± 4.75° after surgery. However, the mean angle in group 2 was 171.47 ± 6.38° before and 165.38 ± 5.23° after surgery. A significant difference between the intercondylar angles before and after surgery was confirmed using the Wilcoxon test in both groups (*p* < 0.001). Furthermore, the alteration in the intercondylar angle after surgery was also significantly different between the two groups (*p* < 0.001).

## 4. Discussion

The purpose of this study was to assess the stability of the post-surgical TMJ position in skeletal Class III cleft and non-cleft patients undergoing bimaxillary surgery under the CASNOS protocol. The CASNOS protocol and the accuracy of the transfer from surgical simulation to actual surgery have been presented in a previous study [12]. All subjects in both groups were diagnosed with skeletal Class III with a retrusive maxilla and improved skeletal pattern after surgery. The mean ANB improved from −6.2° to 2.3° and from −4.9° to 0.9° in groups 1 and 2, respectively. Moreover, the linearity of the mean A-Nv and Pog-Nv in both groups also improved, and the cephalometric data are shown in Table 2. Some studies have discussed the stability of the TMJ position after OGS [15,16,17]. However, few studies have investigated the changes in post-surgical TMJ position in patients with cleft palate. Our study demonstrated the changes in TMJ position before and after two-jaw surgery using 3D image analysis in non-cleft and cleft patients.

After the surgical progress of mandibular setback with BSSO, the proximal segments were moved distally and fixed with the distal segments within the planned surgical occlusion design. Under the fixation force and vector from the temporomandibular ligaments, condyle head rotation may occur [15]. Several studies have examined changes in the condylar axis after mandibular osteotomies, and the condylar axis was shown to be rotated inward in the axial view after BSSO [15,16,17]. Our study revealed significant alterations between the angle of the lateral condyles before and after osteotomy in groups 1 (2.33 ± 1.34°) and 2 (6.30 ± 2.22°). Our results were similar to those of previous studies that reported rotation of condyle heads in skeletal Class III subjects under OGS treatment. [16,17] Although a significant difference in changes in the condyle axis angle after surgery was demonstrated in Ha’s study, the mean change in the condyle axis angle was approximately 5°, which was much more than our data for group 1 showed (2.33°± 1.34°) [16]. The reasons for less rotation of the condyle axis angle in our study might be the application of the CASNOS protocol, which transfers simulation to actual surgery using a navigation system precisely [12], and individual variation of skeletal discrepancy in subjects. Katsumata et al. demonstrated that no obvious condylar axis rotation occurred after BSSO; however, 85.9% of the condyles tended to rotate outward after IVRO [18]. Therefore, different surgical techniques may also influence the post-surgical rotation of the condyle.

Because of the severe discrepancy in bony structures in cleft patients [5], bony remolding and fixation between the proximal and distal segments may result in many problems. Furthermore, soft tissue tension due to scar contracture in patients with cleft often causes postoperative relapse [19]. The reasons might lead to a situation of jawbone relapse. Several studies have reported moderate to high postoperative relapse rates in patients with cleft palate [7,8,9]. Most of these studies focused on relapse of the maxilla after surgery and demonstrated that postoperative relapse at point A was from 20% to 40% horizontally and >50% vertically [7,8]. However, rarely have studies discussed the instability of the TMJ and provided data relating to TMJ position discrepancy and alteration of intercondylar angle postoperatively in cleft patients, which might influence postoperative relapse. Our research presents a significantly greater discrepancy (*p* < 0.001) of the most lateral and medial condylar points and more alteration of the intercondylar angle (*p* < 0.001) in group 2 compared with group 1 (Table 3 and Table 4), which confirmed the significant changes of TMJ positions postoperatively in cleft patients. In cleft patients, the existence of soft tissue scarring caused by numerous craniofacial surgeries on the lip and palate from childhood to adolescence might restrict the jawbone’s movement during surgery and pull the maxilla back toward the original position postoperatively [20]. The high vertical relapse rate (65%) at the maxilla postoperatively was reported by Chua et al. (2010) [7], and counterclockwise rotation of the mandible can occur due to vertical relapse of the maxilla, which might induce a change in the TMJ position postoperatively [11]. These reasons may explain the more significant changes in TMJ position and intercondylar angle in cleft subjects after OGS.

Many factors are involved in immediate condylar displacement. Indirect trauma to the condyle by manipulation of the proximal segment during surgery may cause posterior condylar displacement, and postoperative intra-articular edema may cause inferior condylar displacement. In addition, unpredictable variation in immediate condylar displacement may occur due to condyle displacement out of the fossa or bony interference between the proximal and distal segments during surgery [4]. Through OGS, the muscle relaxants under general anesthesia may induce condyle sag. The TMJ tends to move back to its original position under the force of the masticatory muscles and strain of the TMJ after intermaxillary fixation removal [21]. Sanromán et al. (1997) demonstrated that the condyle sag might be the consequence of many factors, including intra-articular edema, muscle tone, and the new position of the rotated or tilted proximal segment associated with the distal segment, contributing to postoperative instability of the TMJ [4]. In our study, significant changes in condyle position were observed in both groups. The second CT scan was obtained six months postoperatively when the recovery of masticatory function had already taken place. The results of our research correspond to those of Harris’ study. The authors demonstrated that most condylar displacements were noted in their cases, and the condyles were displaced medially, posteriorly, superiorly, and angled medially 2 months after BSSO advancement [22]. However, Chen et al. (2013) reported that the condyle tended to move posteroinferiorly with surgery, but recovery toward the original position was found three months after surgery. Subsequently, the condyle position remained stable and in the centric position of the glenoid fossa throughout the year [21]. The different results between Chen’s and our studies might be due to the dissimilar method design in research. In Chen’s study, the authors investigated the anterior, superior, and posterior spaces of the glenoid fossa in the sagittal view through surgery, and the relationship between the TMJ and glenoid fossa represents indirect changes in TMJ position. Our research studied selected anatomical points of the TMJ in the axial view before and after surgery as direct position changes of the TMJ.

The limitations of this study are the sample size and the long-term investigation period. Additional samples with different types of surgical modalities and fixation techniques were used to assess the stability during OGS. Previous studies have shown that the condyle position remained stable one year after OGS [21]. Therefore, long-term follow-up of the stability after OGS in cleft patients requires further evaluation. Moreover, 3D images gathered immediately after surgery could be considered for further assessment of short-term and long-term changes in TMJ position.

## 5. Conclusions

Based on the present study, significant changes in postoperative TMJ position were confirmed in both the non-cleft and cleft groups. A comparison of these two groups revealed significant postoperative discrepancies in the TMJ and more changes in the intercondylar angle after surgery in the cleft group. This finding may be a reason leading to greater postoperative instability in cleft patients compared with skeletal Class III non-cleft patients.

## Figures and Tables

**Figure 1 jpm-12-01437-f001:**
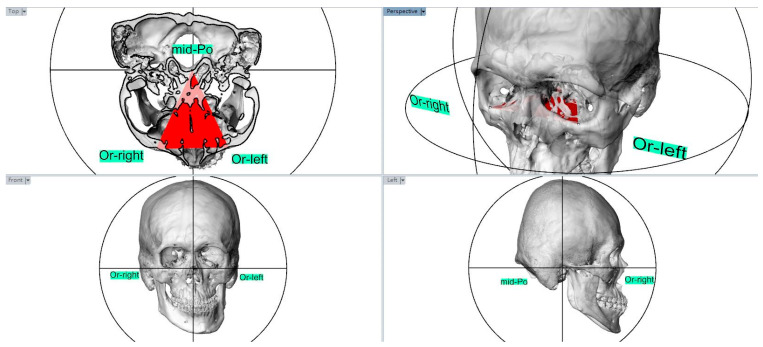
The head orientation was paralleled to the Frankfort horizontal (FH) line as the horizontal reference, and the FH plane was constructed with a landmark of middle point of both Porion (mid-Po) and bilateral Orbitle (Or-right and Or-left).

**Figure 2 jpm-12-01437-f002:**
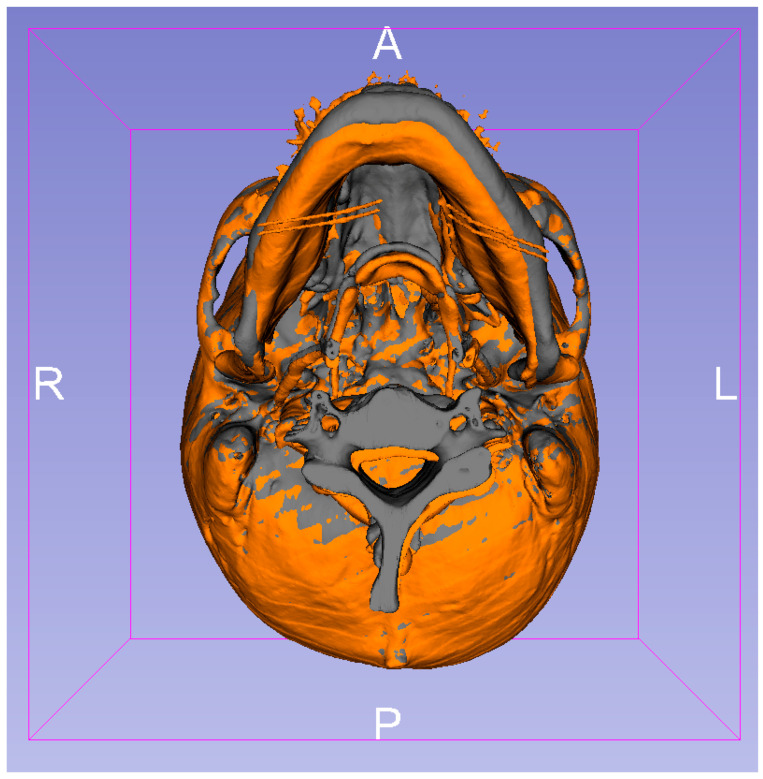
The superimpositions of presurgical (T1, gray color) and post-surgical craniofacial area (T2, orange color) were registered to the cranial base using a voxel-based registration algorithm (inferior view).

**Figure 3 jpm-12-01437-f003:**
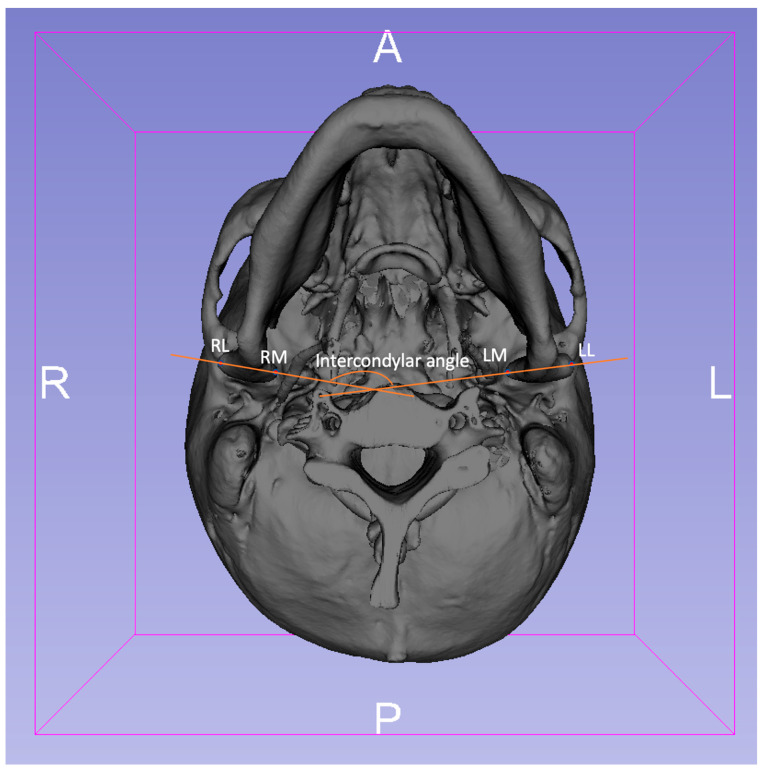
The preoperative (T1) 3D imaging of the inferior view of the presurgical mandible and cranial base. RL (right), LL (left)—most lateral point of the condyle; RM (right), LM (left)—most medial point of the condyle were identified. The cutting angle between the axes (RL-LM and RM-LL), also called the intercondylar angle, was calculated and measured.

**Figure 4 jpm-12-01437-f004:**
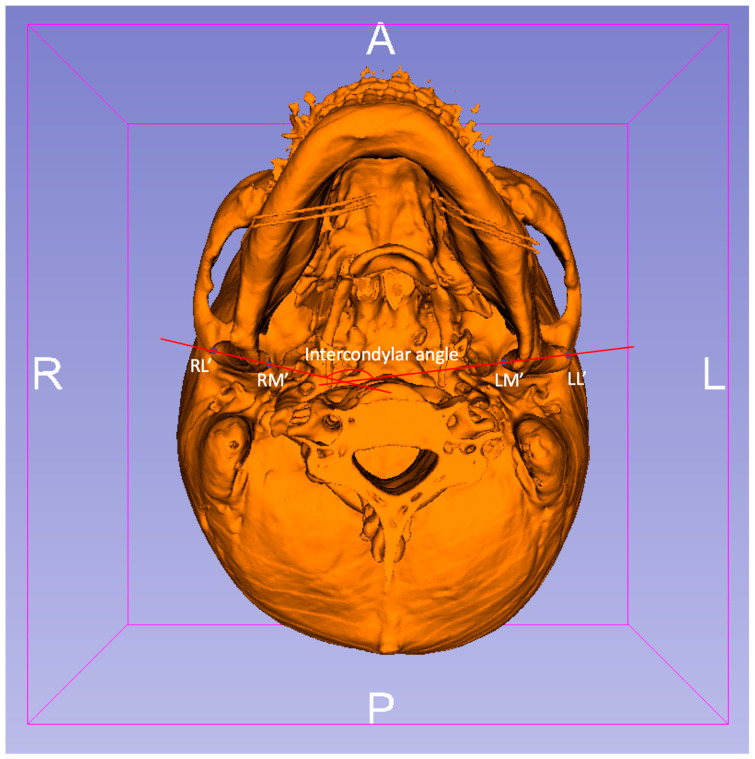
The postoperative (T2) 3D imaging of the inferior view of the presurgical mandible and cranial base. RL’ (right), LL’ (left)—most lateral point of the condyle, RM’ (right), LM’ (left)—most medial point of the condyle were identified. The cutting angle between the axes (RL’-LM’ and RM’-LL’), also called the intercondylar angle, was calculated and measured.

**Table 1 jpm-12-01437-t001:** Distribution of samples (Group 1: skeletal Class III patients; Group 2: skeletal Class III cleft patients with retrusive mid-face) by sex and age.

Group 1			Group 2		
**Sex**	**Amount**	**Mean Age (years)**	**Sex**	**Amount**	**Mean Age (years)**
Male	11	24.9 ± 4.5 years (range: 20~33 years)	Male	14	19.5 ± 2.5 years (range: 18~21 years)
Female	12	23.92 ± 5.2 years (range: 19~36 years)	Female	6	19.2 ± 2.3 years (range: 19~22 years)
Total	23	24.39 ± 4.8 years (range: 19~36 years)	Total	20	20.25 ± 3.8 years (range: 18~22 years)

**Table 2 jpm-12-01437-t002:** Cephalometric measurements taken three weeks before surgery and 2~3 days after surgery.

Group 1			Group 2		
**Measurement**	**Mean Value ± SD** **Pre-Surgery**	**Mean Value ± SD** **Post-Surgery**	**Measurement**	**Mean Value ± SD** **Pre-Surgery**	**Mean Value ± SD** **Post-Surgery**
SNA	79.3 ± 1.5°	83.7 ± 1.3°	SNA	76.1 ± 4.1°	79.7 ± 3.8°
SNB	86.1 ± 1.5°	81.1 ± 1.3°	SNB	80.9 ± 4.1°	78.8 ± 4.1°
ANB	−6.2 ± 1.9°	2.3 ± 1.5°	ANB	−4.9 ± 3.1°	0.9 ± 2.6°
A-Nv	0.5 ± 1.6 mm	2.5 ± 1.2 mm	A-Nv	−7.1 ± 4.4 mm	−2.6 ± 4.2 mm
Pog-Nv	10.7 ± 3.7 mm	1.3 ± 0.6 mm	Pog-Nv	−1.8 ± 10.8 mm	−5.9 ± 7.5 mm

S: Sella; N: Nasion; Point A: Subspinale; Point B: Supramentale; SNA: Sella-Nasion-Point A angle; SNB: Sella-Nasion-Point B angle; ANB: Point A-Nasion-Point B angle; Nv: The line goes through Nasion and is perpendicular to the FH plane; Pog: Pogoion; A-Nv: The distance from point A to the Nv line; Pog-Nv: The distance from Pog to Nv line.

**Table 3 jpm-12-01437-t003:** Discrepancy of the most lateral and medial condylar points between the two groups.

Parameter	Group 1	Group 2		
**Discrepancy of Condylar Point**	**Mean Discrepancy**	**Discrepancy of Condylar Point**	**Mean Discrepancy**	***p*-Value**
RL-RL’	1.23 ± 0.47 mm	RL-RL’	2.53 ± 0.82 mm	<0.001 *
LL-LL’	1.14 ± 0.33 mm	LL-LL’	2.37 ± 0.71 mm	<0.001 *
RM-RM’	1.09 ± 0.23 mm	RM-RM’	2.86 ± 1.02 mm	<0.001 *
LM-LM’	0.95 ± 0.15 mm	LM-LM’	2.42 ± 0.85 mm	<0.001 *

The 3D discrepancy of all parameters showed significant difference between two groups, as confirmed by Mann–Whitney U test (Significance level: * *p* < 0.05).

**Table 4 jpm-12-01437-t004:** The variation and alteration of intercondyar angles in the two groups.

Parameter	Group 1		Group 2		
**Intercondylar Angles**	**Mean Value ± SD**	***p*-value**	**Intercondylar Angles**	**Mean Value ± SD**	***p*-Value**
Preoperative	161.75 ± 5.18°		Preoperative	171.47 ± 6.38°	
Postoperative	159.36 ± 4.75°	<0.001 *	Postoperative	165.38 ± 5.23°	<0.001 *
Alteration of intercondylar angles	2.33 ± 1.34°	<0.001 *	Alteration of intercondylar angles	6.30 ± 2.22°	<0.001 *

The significant difference between intercondylar angles before and after surgery was confirmed in both groups. Additionally, the alteration of intercondylar angle after surgery was also significantly different between the two groups (significance level: * *p* < 0.05).

## Data Availability

The datasets supporting the conclusions of this article are included within the article.

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
