# Peer review of "Comparison of the Post-Surgical Position of the Temporomandibular Joint after Orthognathic Surgery in Skeletal Class III Patients and Patients with Cleft Lip and Palate"

_jpm, 2022, doi:10.3390/jpm12091437_

Round 1
Reviewer 1 Report
Dear Authors, the research is a case-control retrospective study, aimed to highligt differences between the condylar angulation and position after surgery in two groups of Class III patients, with and without cleft lip and palate. The research is well setted, and its strenght and limitation are clearly described. My only concern is on the missing of a sample size calculation, but I noticed that you describe this missing in the limitation of the research.
In my opinion the paper should be published.
Author Response
We appreciate the helpful comment made by the reviewer, which will improve the quality of our manuscript. Thanks for your review and comment.
Reviewer 2 Report
The authors compared the stability of TMJ in orthognathic surgery among non-cleft versus cleft with class III skeletal profile. The use of the term instability and relapse should be clarify further as comparing pre versus post-operative surgical changes only account to the changes that occurred due to the surgical movement itself and not a relapse. Other detailed comments:
1. English correction is needed
2. Proofreading is also necessary with some typing errors
3. Figure 1 cannot be found within the manuscript
4. Either 3A or 3B (only one figure is available) also cannot be found in the manuscript.
5. Results in table 3, regarding the distance between pre versus post op landmarks of the condyle. It is not very clear if the distance measure follows certain plane (anterior posterior distance parallel to a certain axis) or direct measurement between distance. If just a direct measurement, how do we know if the displacement is in vertical or horizontal or depths (X, Y or Z axis).
6. State the P-value in the table 3 and 4
7. It is not clear what the authors meant by “mean displacement”. Does it mean the difference between pre-op and post-op position? Then the displacement is actually caused by the surgical movement, not relapse (or stability). For stability, the displacement must be defined by the “position of condyle landmarks immediately postop versus similar landmarks in few months (6 months in this study)”
Author Response
We are grateful for the excellent suggestions made by the reviewer, which will improve the quality and clarity of the manuscript. The point-by point response to review's comments are demonstrated in the attachment file. Please see the attachment and thanks for your review and comments.
